# Embracing Domain Gradient Conflicts: Domain Generalization Using Domain Gradient Equilibrium

## ABSTRACT

Single domain generalization (SDG) aims to learn a generalizable model from only one source domain available to unseen target domains. Existing SDG techniques rely on data or feature augmentation to generate distributions that complement the source domain. However, these approaches fail to address the challenge where gradient conflicts from synthesized domains impede the learning of domain-invariant representation. Inspired by the concept of mechanical equilibrium in physics, we propose a novel conflict-aware approach named domain gradient equilibrium for SDG. Unlike prior conflict-aware SDG methods that alleviate the gradient conflicts by setting them to zero or random values, the proposed domain gradient equilibrium method first decouples gradients into domain-invariant and domain-specific components. The domain-specific gradients are then adjusted and reweighted to achieve equilibrium, steering the model optimization toward a domain-invariant direction to enhance generalization capability. We conduct comprehensive experiments on four image recognition benchmarks, and our method achieves an accuracy improvement of 2.94% in the PACS dataset over existing state-of-the-art approaches, demonstrating the effectiveness of our proposed approach.

## CCS CONCEPTS

• **Computing methodologies** → **Computer vision; Neural networks; Image representations**.

## KEYWORDS

domain shift, medical image analysis, adversarial domain augmentation, random convolution

## 1 INTRODUCTION

Deep neural networks have achieved remarkable performance on various tasks under standard supervised learning settings where training and test data share the same distribution [14, 21]. However, their performance often deteriorates substantially when the test distribution diverges from the training data because of domain shift, a pervasive challenge that impedes real-world application [29, 32, 56]. For instance, models trained on images from DSLR cameras might underperform on smartphone images because of variations in resolution, noise levels, and other factors. Such domain

shifts are ubiquitous in real-world situations, making it imperative for models to generalize across distributional shifts.

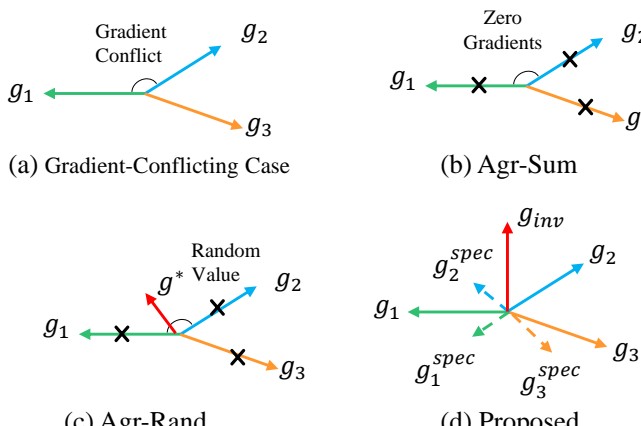

**Figure 1: Comparison of different approaches addressing gradient conflicts in domain generalization. (a) An example of the gradient-conflicting issue. (b) Agr-Sum [25] sets domain gradients to zero, and (c) Agr-Rand [25] assigns them to random values when their signs are inconsistent. (d) Our approach first decouples gradients into domain-invariant and domain-specific components. Subsequently, we adjust the domain-specific components to achieve equilibrium, directing model optimization towards a domain-invariant orientation.**

Many existing works attempt to tackle this problem via unsupervised domain adaptation (UDA) [28, 35, 38] or multiple domain generalization (MDG) [4, 15, 54]. Unsupervised domain adaptation reduces domain gaps by image translation [3, 49] or adversarial training [12, 35]. Unlike UDA, MDG methods extract domain-invariant features from multiple source domains without requiring access to data from the target domain. This enables better generalization to the unseen target domain data. Recent MDG methods learn domain-invariant representations via feature alignment [4, 18] and meta-learning [11, 19] using data aggregated from multiple source domains. However, collecting data from multiple domains is often infeasible due to cost or privacy concerns, particularly in some scenarios such as clinical practice [23, 27], motivating the exploration of using single source domain to learn domain-invariant feature representations. Single domain generalization (SDG) approaches learn generalizable visual representation via self-supervised learning [2, 45] and data augmentation [31, 52]. However, self-supervised schemes tend to be task-specific and tedious in design, while data augmentation approaches rely heavily on the choice of enhancement techniques. Therefore, these methods may not adequately

handle more complex domain distribution shifts, hindering their effectiveness in enhancing model generalization.

Unlike existing SDG methods, our approach strategically utilizes the direction of gradients during the optimization process to cultivate domain-invariant features. Our inspiration stems from recent works that analyze gradient conflicts in multi-task learning (MTL) [5, 22, 36], in which conflicts arise between gradients of different tasks. By contrast, for domain generalization, conflicts emerge across mini-batches of different domains. Such domain gradient interference in each mini-batch can lead models to overfit certain domains, degrading the generalization capability of a model. A recent study [25] introduced the Agr-Sum and Agr-Rand strategies, which address domain gradient conflicts by setting inconsistent gradients to zero and assigning them random values, respectively. However, by discarding or randomizing inconsistent gradients, these strategies may lose pivotal gradient information and limit generalization capabilities. As illustrated in Fig. 1, a comparison between Agr-Sum, Agr-Rand, and the proposed approach is provided.

Our approach addresses the challenges of domain conflicts that arise during domain generalization. These conflicts, particularly under domain shift conditions, are primarily attributed to gradients pulling the model's parameters in conflicting directions, making it difficult for the model to generalize across different domains. To tackle this, we introduce a novel method centered on the gradient equilibrium concept to achieve a gradient consensus among source domains. We start by decomposing the gradients into two components: domain specific and domain invariant. The domain-specific gradients, which often contribute to the conflicts, are then adjusted and reweighted to achieve equilibrium. This ensures that while the orientation and magnitude of domain-specific gradients are recalibrated, the domain-invariant components—those essential for generalization—are retained and emphasized. Our method steers the model optimization toward the domain-invariant direction, fostering more generalizable features. Unlike prior conflict-aware SDG methods [25, 50] that might recalibrate gradients, our strategy is unique in its use of gradient alignment based on the equilibrium principle. In this alignment, the interference from conflicting gradients is reduced, while the domain-invariant knowledge essential for domain generalization is preserved.

Our contributions are summarized as follows:

- We introduce a novel approach, domain gradient equilibrium, focusing on gradient balance to advance single-domain generalization. This method innovatively addresses gradient conflicts, leveraging mechanical equilibrium principles to facilitate learning domain-invariant representations.
- We achieve equilibrium by decomposing gradients into domain-specific and invariant components and strategically adjusting the domain-specific gradients. This process steers the model optimization towards domain invariance, enhancing generalizability.
- We validate our approach on four benchmarks: PACS, VLCS, OfficeHome, and DomainNet, demonstrating superior performance to current SOTAs. This underlines our method's efficacy in enhancing robust domain generalization.

## 2 RELATED WORK

### 2.1 Domain Generalization

Domain generalization has witnessed significant progress, aiming to learn representations that generalize across unseen target distributions [43]. Earlier approaches mainly extracted domain-invariant representations from multi-source domain data [1, 11]. However, the high costs and privacy concerns in some scenarios often make this approach impractical. Recent research has increasingly focused on SDG with only one source domain data available for training. The proposed methods encompass self-supervised learning [2], data augmentation strategies [31, 52], and the use of randomized convolutions [7, 47]. Self-supervised techniques have been employed to extract domain-invariant features [2], while augmentation-based methods attempt to simulate target distributions by applying stacked transformations [52]. However, the design intricacies of self-supervision and the inherent limitations of augmentations pose challenges. Several studies have explored gradient-based strategies in domain generalization. A recent study [25] introduced the Agr-Sum and Agr-Rand consensus strategies to alleviate domain gradient conflicts and improve generalization capability. However, these strategies might discard crucial gradient information by setting inconsistent domain gradients to zero or assigning random values, potentially compromising the model's generalization capabilities.

### 2.2 Gradient Conflicts in Multi-task Learning

Multi-task learning (MTL) [8, 40, 53] is widely used to learn efficient models by sharing parameters across related tasks. However, gradient conflict is a major challenge when simultaneously optimizing multiple tasks, as gradients of different tasks could have opposite directions [22, 36]. Earlier works [5, 13, 17] alleviated the influence of dominant gradients by reweighting task losses using uncertainty-based [17], norm-based [5], and difficulty-based [13] weighting. More recent approaches directly manipulate conflicting gradients for better alignment, such as PCGrad [50] and GradDrop [6]. However, most methods lack convergence guarantees. Recent research has investigated network architecture design to isolate task-specific modules and reduce gradient conflicts, such as Recon [33] and CoNAL [51]. Despite advancements in MTL, they primarily address conflicts between tasks and may not be directly applicable to domain generalization, where conflicts emerge across different domain mini-batches. We propose a domain gradient equilibrium method involving a gradient decomposition strategy, separating gradients into domain-specific and domain-invariant components. By adjusting and reweighting the domain-specific gradients to reach equilibrium, our method capitalizes on gradient conflicts and steers optimization towards learning domain-invariant features, improving the model's generalization ability across various domains.

## 3 METHODOLOGY

Unlike prior conflict-aware SDG methods that set the conflicting gradients as zero or random values [24, 25], we propose a novel approach, domain gradient equilibrium, for enhancing model generalization across domains, as shown in Fig. 2. The proposed approach seeks better gradient updates in which prediction consistency is

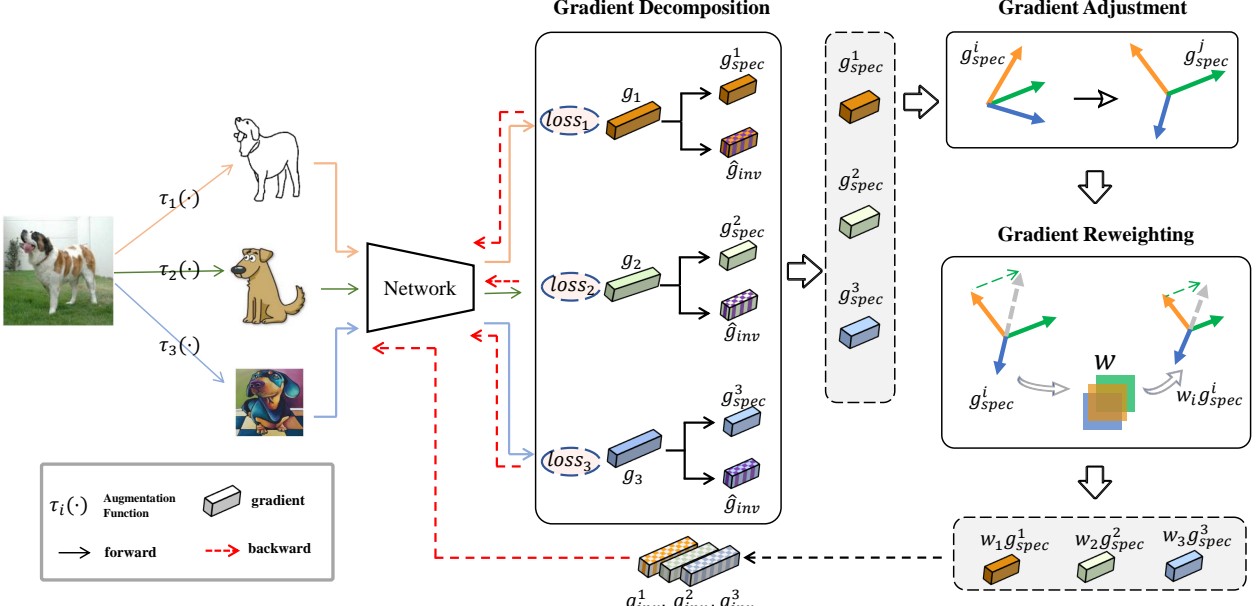

**Figure 2: Overview of the Domain Gradient Equilibrium method. The method initiates by applying multiple augmentations to input images to simulate domain variability. For each augmented instance, the network calculates individual losses and the corresponding gradients. These gradients are decomposed into domain-invariant and domain-specific components. The domain-specific components are subsequently adjusted to maintain substantial angular separation between domains, followed by a reweighting process that equilibrates these domain-specific gradients. Finally, the reweighted domain-specific gradients are used to derive refined domain-invariant gradients, guiding the optimization toward the domain-invariant direction, thereby enhancing the model's generalization capabilities.**

improved across all synthesized source domains by leveraging the conflicting gradients. Inspired by force balance, we employ a gradient balancing strategy to address conflicts in gradient orientations across domains. Specifically, we decompose the gradients into domain-specific and domain-invariant components. We then adjust the angular directions of the domain-specific gradients using gradient projection to ensure distinct separations. Lastly, each domain-specific gradient is reweighted to achieve balance. In this manner, the model optimizes in the direction of the domain-invariant gradient, enhancing the acquisition of generalizable features.

### 3.1 Gradient Decomposition

Our approach to gradient decomposition is similar to the method presented in [37]. While both techniques utilize gradient decomposition, they diverge in strategy and purpose. The method in [37] partitions the gradient of a prior task into two parts: one shared among all previous tasks and another specific to the task at hand to mitigate catastrophic forgetting in continual learning. Conversely, our approach decouples domain gradients into domain-specific and domain-invariant components, targeting a balanced gradient to bolster a model's resilience to domain variations. Specifically, we first computed losses over the synthesized source domains generated by performing data augmentation $\tau(\cdot)$ on the source domain data,

as formulated below:

$$\mathcal{L}_i(\theta) = \sum_{j=1}^{M} \mathcal{L}_{ce}(\tau_i(x_j), y_j; \theta), \tag{1}$$

where $M$ represents the number of the samples for each domain, $\mathcal{L}_{ce}$ is the cross-entropy loss, and $\tau_i(\cdot)$ denotes the $i_{th}$ data augmentation function performed on the source domain $\mathcal{D}_s = \{x_j, y_j\}_{j=1}^{M}$. The data augmentation function $\tau_i(\cdot)$ was implemented using RandAugment [9] with *random_ops* randomly selected data augmentation operations to bolster the diversity of augmentations. Subsequently, the gradient vectors for $K$ domains, $\{g_0, g_1, ..., g_{K-1}\}$, are derived via backpropagation. The updated gradients are subsequently decoupled into domain-invariant and domain-specific components:

$$g = g_{inv} + g_{spec}, \tag{2}$$

where $g_{inv}$ denotes the domain-invariant gradient, and $g_{spec}$ represents the domain-specific gradient. Under the assumption that the domain-invariant gradient exhibits the minimum divergence from each gradient vector [37], the optimization problem can be reformulated as follows:

$$\min_{g_{inv}} \sum_{i=1}^{K} ||g_{inv} - g_i||_2^2, \tag{3}$$

where $|| \cdot ||_2^2$ represents the squared $L_2$ norm. By solving this optimization problem, the coarse domain-invariant gradient $\hat{g}_{inv} =$

$(\mathbf{g}_0 + \mathbf{g}_1 + ... + \mathbf{g}_{K-1})/K$ can be derived, which constitutes the mean of all gradient vectors. Finally, the domain-specific gradient for the $k$-th domain is formulated as:

$$\mathbf{g}_{spec}^k = \mathbf{g}_k - \hat{\mathbf{g}}_{inv} \tag{4}$$

By decomposing each gradient vector into domain-invariant and domain-specific components, the domain-specific gradients can be individually adjusted to ensure the retention of the domain-invariant gradient direction throughout model optimization. Such adjustments mitigate cross-domain discrepancies and facilitate the learning of generalizable representations.

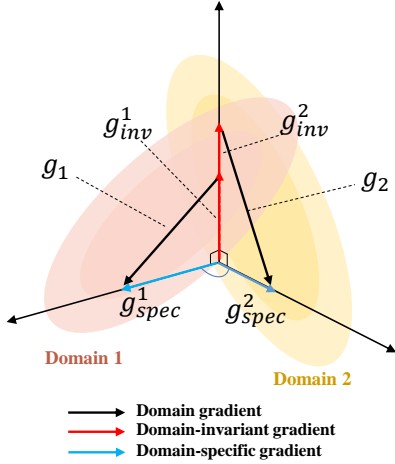

**Figure 3: Illustration of gradient decomposition for two distinct domains. For each domain, the gradient is decomposed into domain-specific and domain-invariant components.**

### 3.2 Gradient Adjustment

Existing MTL approaches mainly focus on identifying and depressing conflicting gradients to resolve gradient conflicts [5, 22, 36]. Unlike these methods, we propose embracing gradient conflicts by guaranteeing sufficiently large angles between domain-specific gradients. The rationale is to harness the balancing principle for attenuating the interference of domain-specific gradients, thereby optimizing the model toward domain-invariant gradient orientation and enhancing model generalization capability. Specifically, let $\mathbf{g}_{spec}^i$ denote the domain-specific gradient for domain $i$, and let $\mathbf{g}_{spec}^j$ represent the domain-specific gradient for domain $j$. The adjustment procedures are described as follows:

(1) Compute the cosine similarity between $\mathbf{g}_{spec}^i$ and $\mathbf{g}_{spec}^j$.

(2) If the similarity is above a given threshold $\alpha$, project the domain-specific gradient $\mathbf{g}_i^{spec}$ onto the normal plane of the other domain-specific gradient $\mathbf{g}_j^{spec}$:

$$\mathbf{g}_{spec}^i = \mathbf{g}_{spec}^i + \frac{\mathbf{g}_{spec}^i \cdot \mathbf{g}_{spec}^j}{||\mathbf{g}_{spec}^j||_2^2} \mathbf{g}_{spec}^j, j \neq i \tag{5}$$

This step seeks to decrease the similarity among domain-specific gradients, ensuring that the angle between these domain-specific

gradients is increased to achieve equilibrium. The threshold $\alpha$ is empirically set at 0.5, considering similarity ranges [0,1].

(3) Repeat this process by modifying $\mathbf{g}_{spec}^i$ against the gradients of other domains $\mathbf{g}_{spec}^j$ in the current batch.

### 3.3 Gradient Reweighting

After adjusting the domain-specific gradients, assigning appropriate weights to each gradient is essential to achieve optimal equilibrium. Therefore, we aim to find optimal weights $\mathcal{W} = \{w_1, w_2, ..., w_k\}$ to linearly combine multiple domain-specific gradients into a zero vector:

$$w_1\mathbf{g}_{spec}^1 + w_2\mathbf{g}_{spec}^2 + ... + w_k\mathbf{g}_{spec}^k = \mathbf{0}. \tag{6}$$

With these weights, we can mitigate the conflicts of domain-specific gradients and steer the model optimization toward the domain-invariant direction. To this end, we formulate the following objective function:

$$\min_{\mathbf{w}_k} ||w_1\mathbf{g}_{spec}^1 + w_2\mathbf{g}_{spec}^2 + ... + w_k\mathbf{g}_{spec}^k||_2^2$$
$$s.t. \quad 0 \leq w_k \leq 1. \tag{7}$$

Using gradient descent, the weight update rules are as follows:

$$w_1^{(t+1)} = \max\left(0, \min\left(1, w_1^{(t)} - \eta\frac{\partial L}{\partial w_1}\right)\right),$$
$$\vdots$$
$$w_k^{(t+1)} = \max\left(0, \min\left(1, w_k^{(t)} - \eta\frac{\partial L}{\partial w_k}\right)\right), \tag{8}$$

where $\eta$ is the learning rate. The iterative reweighting process is designed to mitigate the influence of domain-specific gradients. This ensures that the model optimization is consistently aligned with the domain-invariant gradient direction, facilitating the learning of transferable representations. Finally, for each domain $k$, the refined domain-invariant gradient $\mathbf{g}_{inv}^k$ can be derived from the reweighted domain-specific gradients, which can be formulated as follows:

$$\mathbf{g}_{inv}^k = \mathbf{g}_k - w_k\mathbf{g}_{spec}^k, \tag{9}$$

where $\mathbf{g}_k$ is the total gradient for domain $k$, $\mathbf{g}_{spec}^k$ is the domain-specific gradient for domain $k$, and $w_k$ is the weight assigned to the domain-specific gradient. For a simple comparison, the number of augmented source domains $K$ is set to 3. The complete procedure of the domain gradient equilibrium approach is outlined in **Algorithm 1**.

During the training process, we sequentially use the domain-invariant gradient $\mathbf{g}_{inv}^k$ of each domain to update the model parameters. This approach ensures that the model is optimized towards a domain-invariant direction for each domain while reducing the influence of domain-specific information.

## 4 EXPERIMENTS

We evaluate the effectiveness of the proposed approach for SDG using four widely used image recognition benchmarks, namely, PACS [20], VLCS [39], OfficeHome [42], and DomainNet [30]. PACS [20] is a widely used benchmark for domain generalization with four domains: Photo (P), Art Painting (A), Cartoon (C), and Sketch (S).

---

**Algorithm 1** Domain Gradient Equilibrium

---

1: **Input:** Training data from $N$ source domains $D = \{D_1, D_2, \ldots, D_N\}$, learning rate $lr$
2: **Output:** Updated model parameters $\theta$
3: Initialize model parameters $\theta$, weights $w_1, w_2, \ldots, w_K$
4: **for** $t = 1$ to $T$ **do**
5:     Sample a mini-batch of size $B$ from $D$
6:     Compute losses over the augmented domains using Eq. 1.
7:     Compute $\hat{g}_{inv}$ using Eq. 3.
8:     Compute $g_{spec}^k$ using Eq. 4.
9:     Adjust the domain-specific gradients using Eq. 5.
10:     Update weights using Eq. 8.
11:     Compute $g_{inv}^k$ using Eq. 9.
12:     Update model parameters $\theta$
13: **end for**
14: **return** $\theta$

---

VLCS [39], a commonly adopted benchmark for domain generalization with four different domains: VOC2007 (V), LabelMe (L), Caltech101 (C), and SUN09 (S). OfficeHome [42] is a cross-domain object recognition benchmark that contains four domains: Art (A), Clipart (C), Product (P), and Real-World (R). DomainNet [30] is a large-scale benchmark dataset for domain adaptation and generalization, consisting of six domains: Clipart (C), Infograph (I), Painting (P), Quickdraw (Q), Real (R), and Sketch (S). Experiments were conducted under both single and multiple source domain settings to comprehensively evaluate the effectiveness of our method.

## 4.1 Single Domain Generalization on Image Recognition

**Setup and Implementation Details:** In our experiments, we utilized the ResNet-18 [14] as the backbone pre-trained on ImageNet [10] for all compared methods. For optimization, we employed the SGD optimizer with a momentum of 0.9 and a learning rate of 5e-4 for all tasks. The batch size was set to 10. Following the protocol in [46], we divided the source domain dataset into training and validation subsets. We chose the model with the best validation performance for reporting results. For the experimental setup, one domain was designated as the source domain for training, and the remaining domains were used as target domains for testing. For comparison, we used the Empirical Risk Minimization [41] as the baseline approach, which directly employs a vanilla strategy to train the source model. All experiments were conducted using PyTorch 1.10 on an NVIDIA A40 GPU.

**Experimental Results:** We first conducted our experimental analysis on the PACS dataset, as demonstrated in Table 1. From this result, we observed that while most data augmentation methods enhance model performance beyond the baseline, our proposed method consistently surpassed these techniques. Subsequently, we extended our assessment to the OfficeHome dataset, as shown in Table 2. Aligning with our findings from the PACS dataset, our method yielded consistent performance elevation. Further experiments were conducted on the VLCS dataset, as presented in Table 3. The domain shift in PACS and OfficeHome primarily originates from stylistic variations, whereas in VLCS, it is due to background

**Table 1: Single domain generalization accuracy (%) on PACS. Each column denotes the source domain.**

| Methods | A | C | P | S | Avg. |
|---|---|---|---|---|---|
| Baseline | 68.48 | 71.68 | 36.76 | 40.18 | 54.28 |
| Mixup [48] | 67.60 | 67.13 | 51.59 | 47.04 | 58.34 |
| SelfReg [18] | 74.53 | 67.99 | 42.60 | 55.31 | 60.11 |
| RandConv [47] | 73.51 | 70.57 | 40.22 | 53.80 | 59.53 |
| Pro-RandConv [7] | 69.85 | 72.66 | 42.57 | 55.95 | 60.26 |
| Fish [34] | 67.73 | 68.56 | 44.86 | 60.01 | 60.29 |
| SagNet [26] | 73.79 | 71.39 | 50.75 | 49.77 | 61.43 |
| Arg-Rand [25] | 68.95 | 64.35 | 45.40 | 35.15 | 53.46 |
| Arg-Sum [25] | 74.88 | 76.73 | 55.28 | 58.22 | 66.28 |
| GSAM [57] | 68.29 | 67.88 | 38.53 | 29.81 | 51.13 |
| SAGM [44] | 65.94 | 70.10 | 42.14 | 53.65 | 57.96 |
| Mixstyle [55] | 72.87 | 75.42 | 43.14 | 45.70 | 59.28 |
| PCGrad [50] | 77.31 | **78.52** | 54.73 | 58.37 | 67.23 |
| Ours | **79.08** | 78.33 | **62.74** | **60.53** | **70.17** |

Results are averaged over five runs, with the best results bolded and the second best underlined.

and viewpoint diversities. Existing methods were less effective on VLCS, achieving marginal gains. Despite this, our method yielded a significant improvement, elevating the baseline accuracy from 53.17% to 68.11%. We also conducted experiments on the Domain-Net dataset, as shown in Table 4. Our method achieved the highest average accuracy of 27.75%, outperforming strong baselines such as Mixstyle and PCGrad by 2.52% and 0.76%, respectively. These results, along with our findings on PACS, OfficeHome, and VLCS, demonstrate the effectiveness and versatility of our method in enhancing model robustness and generalization capability across diverse domain shifts.

**Table 2: Single domain generalization accuracy (%) on Office-Home. Each column denotes the source domain.**

| Methods | A | C | P | R | Avg. |
|---|---|---|---|---|---|
| Baseline | 42.74 | 39.31 | 39.86 | 52.50 | 43.60 |
| Mixup | 43.69 | 40.68 | 38.56 | 52.12 | 43.76 |
| SelfReg | 50.84 | 44.65 | 42.71 | 56.83 | 48.76 |
| RandConv | 45.79 | 40.20 | 37.45 | 52.87 | 44.08 |
| Pro-RandConv | 46.06 | 39.20 | 40.41 | 49.46 | 43.78 |
| Fish | 45.10 | 39.74 | 36.67 | 52.09 | 43.40 |
| SagNet | 49.84 | 43.01 | 41.42 | 55.61 | 47.47 |
| Arg-Rand | 42.18 | 44.67 | 53.15 | 57.45 | 49.37 |
| Arg-Sum | 49.23 | 44.96 | 44.76 | 55.05 | 48.50 |
| GSAM | 28.97 | 41.40 | 41.72 | 51.50 | 40.90 |
| SAGM | 47.42 | 43.27 | 41.36 | 54.97 | 46.76 |
| Mixstyle | 51.19 | 48.73 | 46.85 | 55.88 | 50.66 |
| PCGrad | 49.91 | 46.61 | 46.11 | 56.31 | 49.74 |
| Ours | **51.77** | **49.09** | **48.59** | **57.82** | **51.81** |

**Table 3: Single domain generalization accuracy (%) on VLCS. Each column denotes the source domain.**

| Methods | C | L | S | V | Avg. |
|---|---|---|---|---|---|
| Baseline | 27.89 | 46.82 | 65.64 | 72.34 | 53.17 |
| Mixup | 31.85 | 43.09 | 62.51 | 75.40 | 53.21 |
| SelfReg | 34.50 | 51.34 | 57.37 | 75.40 | 54.65 |
| RandConv | 35.90 | 58.89 | 59.22 | 76.46 | 57.62 |
| Pro-RandConv | 44.23 | 49.52 | 60.97 | 73.44 | 57.04 |
| Fish | 36.31 | 66.19 | 65.24 | 75.28 | 60.76 |
| SagNet | **73.79** | **71.39** | 50.75 | 49.77 | 61.43 |
| Arg-Rand | 54.96 | 60.42 | 61.96 | 76.99 | 63.58 |
| Arg-Sum | 56.82 | 64.69 | 65.08 | 77.01 | 65.90 |
| GSAM | 53.25 | 61.62 | 60.98 | 75.46 | 62.83 |
| SAGM | 28.17 | 52.58 | 65.97 | 76.36 | 55.77 |
| Mixstyle | 58.44 | 64.74 | 65.57 | 76.35 | 66.27 |
| PCGrad | 56.26 | 67.46 | 65.34 | 76.83 | 66.47 |
| Ours | 60.76 | 65.96 | **66.56** | **79.17** | **68.11** |

**Table 4: Single domain generalization accuracy (%) on DomainNet. Each column denotes the source domain.**

| Methods | C | I | P | Q | S | Avg. |
|---|---|---|---|---|---|---|
| Baseline | 31.88 | 10.44 | 31.80 | 2.82 | 21.26 | 19.65 |
| Mixup | 33.27 | 10.74 | 32.84 | 3.13 | 23.86 | 20.77 |
| SelfReg | 32.38 | 10.89 | 32.65 | 3.84 | 24.06 | 20.77 |
| RandConv | 31.88 | 10.44 | 31.80 | 2.82 | 21.26 | 19.65 |
| Pro-RandConv | 34.41 | 10.84 | 32.22 | 4.14 | 22.98 | 20.92 |
| Fish | 33.01 | 10.39 | 32.66 | 2.94 | 22.15 | 20.23 |
| SagNet | 33.84 | 11.07 | 32.71 | 3.11 | 22.77 | 20.70 |
| Agr-Sum | **43.36** | 12.28 | 40.13 | **6.71** | **32.48** | 26.99 |
| GSAM | 34.59 | 10.40 | 32.79 | 2.72 | 20.85 | 20.27 |
| SAGM | 34.47 | 11.13 | 33.40 | 3.12 | 23.68 | 21.16 |
| Mixstyle | 40.36 | 12.80 | 39.41 | 4.48 | 29.11 | 25.23 |
| PCGrad | 42.34 | **13.96** | **41.32** | 5.86 | 31.45 | 26.99 |
| Ours | 41.34 | 12.15 | 38.41 | 6.10 | 30.75 | **27.75** |

The 'R' dataset is selected as the target domain for testing, while the remaining datasets are employed as the source domains for training.

## 4.2 Multiple Domain Generalization on Image Recognition

**Setup and Implementation Details:** Our experimental setup for MDG in image recognition is similar to the experimental setup used in SDG, utilizing the same image datasets: PACS [20], VLCS [39], and OfficeHome [42]. The main difference lies in the validation process for MDG, where one domain is specifically chosen as the target for validation while the others serve as source domains for training. Regarding implementation, as our model backbone, we continue with ResNet-18 [14] pre-trained on ImageNet [10]. We employ the SGD optimizer with a momentum of 0.9 and a batch size of 10. The learning rate is set to 5e-4 for all tasks.

**Experimental Results:** The comparative analysis of various methods on PACS, VLCS, and OfficeHome datasets, as shown in Table 5. Data augmentation-based, self-supervised learning-based,

and conflict-aware approaches exhibit superior performance to the baseline, indicating the effectiveness of domain generalization techniques. Our domain gradient equilibrium approach, in particular, shows consistently enhanced performance across these datasets. For instance, it improves the accuracy of the OfficeHome dataset from the baseline of 57.83% to 66.84%. This improvement underscores our method's capability in addressing domain generalization challenges across various datasets.

**Table 5: Multiple source domain generalization accuracy (%) on three datasets: PACS, VLCS, and OfficeHome.**

| Methods | PACS | VLCS | OfficeHome | DomainNet | Avg. |
|---|---|---|---|---|---|
| Baseline | 76.72 | 70.74 | 57.83 | 49.30 | 63.65 |
| MixUp | 77.25 | 74.10 | 59.99 | 51.27 | 65.65 |
| SelfReg | 71.54 | 73.85 | 62.06 | 52.17 | 64.90 |
| RandConv | 76.04 | 71.62 | 59.28 | 52.43 | 64.84 |
| Pro-RandConv | 78.62 | 73.06 | 58.82 | 52.91 | 65.85 |
| Fish | 76.29 | 74.10 | 59.63 | 52.80 | 65.71 |
| SagNet | 77.00 | 73.68 | 57.63 | 52.41 | 65.18 |
| Agr-Rand | 85.01 | 78.29 | 62.45 | 44.68 | 67.61 |
| Agr-Sum | 73.53 | 72.85 | 49.37 | 52.53 | 62.07 |
| GSAM | 79.55 | 73.68 | 62.43 | 50.34 | 66.50 |
| SAGM | 79.01 | 75.16 | 59.44 | 52.65 | 66.57 |
| Mixstyle | 83.83 | 76.63 | 63.47 | 52.66 | 69.15 |
| PCGrad | 85.79 | **78.32** | 64.89 | 53.72 | 70.68 |
| Ours | **85.90** | 77.98 | **66.84** | **54.15** | **71.22** |

The last column, "Avg.," represents the average performance across the three datasets. Results are averaged over five runs, with the best results bolded and the second best underlined.

**Table 6: Ablation study on the classification tasks. ✓ denotes the enabled component, while × denotes the disabled component.**

| Grad Adjust | Grad Weight | Dataset | | |
|---|---|---|---|---|
| | | VLCS | OfficeHome | PACS |
| × | × | 66.75 | 47.04 | 68.84 |
| ✓ | × | 66.86 | 48.25 | 69.20 |
| ✓ | ✓ | 68.11 | 50.34 | 70.17 |

## 4.3 Ablation Study

**Contribution of each component.** In our ablation study, detailed in Table 6, we assessed the impact of individual components within our domain gradient equilibrium method. This study involved comparing our complete method against variations featuring different combinations of the gradient adjustment and gradient weighting modules across various domain datasets. The results reveal that jointly employing gradient adjustment and gradient weighting modules yields the most favorable results in all datasets. This observation suggests the integral role these components play in enhancing domain generalization. Notably, activating both modules leads to

marked improvements in performance on the VLCS, OfficeHome, and PACS datasets. Such enhancements indicate our method's efficacy in facilitating the learning of more generalizable features for classification tasks.

**Parameter sensitivity.**

To evaluate the impact of the *random_ops* parameter on our method's performance, we conducted experiments on PACS, VLCS, and OfficeHome datasets. As shown in Fig. 4, the optimal value of *random_ops* varies across datasets. For PACS, the accuracy peaks at *random_ops* = 4 (70.17%), while VLCS shows stable performance across different *random_ops* settings. OfficeHome experiences a decline in accuracy as *random_ops* increases. These results suggest that the optimal level of augmentation complexity depends on the specific dataset and its domain variations. Based on our experiments, a *random_ops* value of 2 provides an optimal balance, yielding effective generalization performance across all datasets. We also investigated the impact of the similarity threshold $\alpha$ on our method's performance using the PACS dataset. As illustrated in Fig. 5, the accuracy initially improves as $\alpha$ increases from 0.1 to 0.5, attaining a maximum of 70.39% at $\alpha = 0.5$. Further increasing $\alpha$ results in a gradual decline in performance, with accuracy decreasing to 69.96% at $\alpha = 0.9$. These results indicate that a moderate similarity threshold ($\alpha = 0.5$) yields the optimal generalization performance. An excessively low threshold may lead to excessive adjustment of domain-specific gradients, while an overly high threshold may result in insufficient equilibrium of domain-specific gradients across domains.

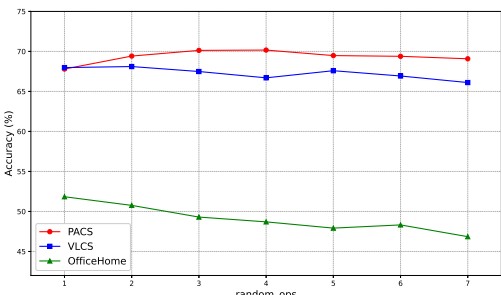

**Figure 4: Classification accuracy across three datasets (PACS, VLCS, and OfficeHome) with varying *random_ops* settings.**

**Impact of the number of augmented sources:** We conducted experiments on the PACS dataset with varying K (the number of augmented sources) values: 2, 3, and 4. As shown in Table 7, we observe a clear upward trend in model performance as K increases. The average accuracy improves from 69.86% with K=2 to 70.17% with K=3, and further to 70.42% with K=4. These results demonstrate the positive impact of increasing the number of augmented sources on domain generalization performance. The ablation study highlights the importance of considering multiple augmented sources to enhance the model's ability to learn domain-invariant features. The consistent improvement in accuracy underscores the effectiveness of our approach in promoting domain generalization.

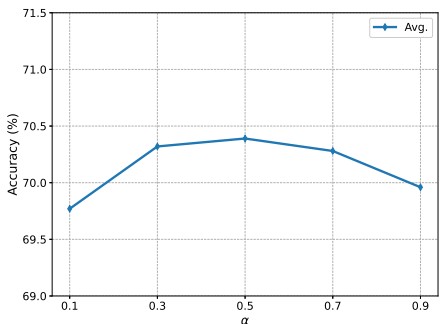

**Figure 5: Classificaiton accuracy over the PACS dataset with varying $\alpha$ settings.**

**Table 7: Ablation study on the impact of varying the number of augmented sources (K) on domain generalization performance using the PACS dataset.**

| Setting | A | C | P | S | Avg. |
|---------|-------|-------|-------|-------|-------|
| K=2 | 77.15 | 78.30 | 62.29 | 61.71 | 69.86 |
| K=3 | 79.08 | 78.33 | 62.74 | 60.53 | 70.17 |
| K=4 | 78.13 | 78.92 | 62.05 | 62.56 | 70.42 |

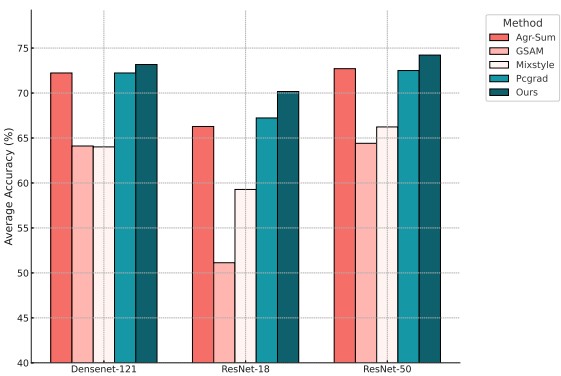

**Figure 6: Performance comparison of the methods using different backbones including the ResNet-18 [14], ResNet-50 [14], and Densenet [16] models on the PACS dataset.**

## 4.4 Further Analysis

**Evaluation on different backbones.** We conducted experiments to assess the effectiveness of our domain gradient equilibrium method across different architectures, including ResNet-18, ResNet-50, and Densenet, on the PACS dataset. The results, as presented in Figure 6, demonstrate that our method consistently outperforms other state-of-the-art methods such as Agr-Sum, Mixstyle, GSAM, and PCGrad across all architectures. Specifically, our method achieved an average performance of 70.17% with ResNet-18, 74.22% with ResNet-50, and 73.17% with Densenet-121 across the datasets, showcasing its robustness and effectiveness in enhancing the generalization capability of models across various architectures.

**Effectiveness of gradient equilibrium beyond simple gradient averaging:** We conducted experiments to evaluate the effectiveness of our domain gradient equilibrium method in comparison with a direct approach that utilizes the average domain gradient for optimization. The experiments were conducted under SDG and MDG settings, as presented in Table 8. The results indicate that simply using the average domain gradient (ours_gradAvg) fails to fully capture domain-invariant representations. It becomes especially evident in the SDG setting in which the performance disparity between ours and ours_gradAvg is more noticeable. For instance, on the PACS dataset under SDG, our method achieved a performance of 70.17%, a significant improvement over the 67.23% achieved by the ours_gradAvg method. Moreover, under the MDG setting, our approach continuously outperformed the ours_gradAvg method across all datasets. This result suggests that the domain-invariant gradient refined by our method is more closely aligned with the actual domain-invariant gradient, thereby enhancing the model to learn better domain-invariant representations.

**Table 8: Comparison of model performance using domain gradient equilibrium (ours) versus direct average domain gradient optimization (ours_gradAvg) under SDG and MDG settings on the PACS, VLCS, and OfficeHome datasets.**

| Settings | Methods | PACS | VLCS | OfficeHome | Avg. |
|---|---|---|---|---|---|
| MDG | ours_gradAvg | 85.01 | 76.92 | 64.85 | 75.59 |
| | ours | 85.90 | 77.98 | 66.84 | 76.91 |
| SDG | ours_gradAvg | 67.23 | 67.75 | 50.33 | 61.77 |
| | ours | 70.17 | 68.11 | 51.81 | 63.36 |

**Impact of data augmentation on domain generalization:** To investigate the effectiveness and versatility of our proposed gradient equilibrium method, we conducted experiments on the PACS dataset using various data augmentation techniques. As shown in Table 9, GE consistently improves the performance across all augmentation settings, demonstrating its ability to enhance domain generalization. Notably, GE combined with RandAugment achieves the highest average accuracy of 70.17%, outperforming the baseline by 3.46%. Furthermore, even with simple augmentations like random rotation and brightness augmentation, GE boosts the average accuracy by 4.28% and 3.70%, respectively. These results suggest that the effectiveness of GE is not limited to specific augmentations but can be applied in conjunction with various techniques to improve domain generalization performance. The consistent improvements achieved by GE across different settings highlight its potential as a general-purpose method for enhancing the robustness and transferability of models in domain generalization tasks.

To further verify that our method's improvements extend beyond the augmentation process, we compared our approach with the vanilla baseline across different numbers of random augmentation operations (*random_ops*). As shown in Figure 7, our method consistently outperforms the vanilla baseline at each level of augmentation, with the performance gap widening as the number of *random_ops* increases. The consistent improvements over the baseline across various augmentation settings underscore the robustness

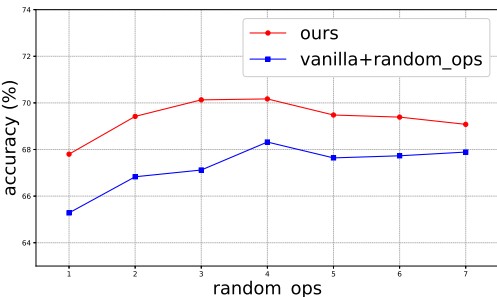

**Figure 7: Comparison of model accuracy with varying** *random_ops* **settings.**

**Table 9: Comparison of the performance of different data augmentation methods on the PACS dataset.**

| Methods | A | C | P | S | Avg. |
|---|---|---|---|---|---|
| baseline+rotation | 68.77 | 70.98 | 37.09 | 48.77 | 56.40 |
| GE+rotation | 74.94 | 72.33 | 42.58 | 52.86 | 60.68 |
| baseline+brightness | 71.73 | 73.25 | 40.69 | 51.57 | 59.31 |
| GE+brightness | 76.82 | 75.99 | 44.79 | 54.43 | 63.01 |
| baseline+rotation+brightness | 68.10 | 71.47 | 38.80 | 51.31 | 57.42 |
| GE+rotation+brightness | 72.98 | 74.74 | 44.14 | 57.05 | 62.23 |
| baseline+RandAugment | 74.12 | 76.26 | 56.22 | 60.23 | 66.71 |
| GE+RandAugment (Ours) | 79.08 | 78.33 | 62.74 | 60.53 | 70.17 |

'GE' represents 'Gradient Equilibrium'. Random rotation augmentation is applied with the rotation angle randomly selected from the range [-30, 30] degrees, and random brightness augmentation is performed with the brightness adjustment factor randomly chosen from the range [0.7, 1.3].

and effectiveness of our approach in enhancing domain generalization performance beyond simple data augmentation. These findings provide strong evidence that our method's benefits are not primarily derived from the augmentation process but from its innovative design and ability to capture domain-invariant features.

# 5 CONCLUSION

We introduced domain gradient equilibrium, a novel approach that embraces domain gradient conflicts to enhance domain generalization. Unlike previous methods that attempted to mitigate gradient conflicts through gradient modification, our method acknowledges and leverages these conflicts. By decomposing gradients into domain-specific and domain-invariant components, the domain-specific gradients are carefully adjusted and reweighted to achieve equilibrium, steering the model optimization toward a domain-invariant direction. Our experiments across various datasets and architectures in the image recognition task demonstrate the approach's robustness and effectiveness. In future work, we will explore our methods with more architectures, such as vision transformers, and expand to more vision tasks.

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
