# OpenReview forum: "Embracing Domain Gradient Conflicts: Domain Generalization Using Domain Gradient Equilibrium"
_acmmm.org/ACMMM/2024/Conference — MM2024 Poster_

### Official Review · Reviewer_zCM3 · 2024-05-13

**Rating:** 3
**Confidence:** 4

**Summary:**

This paper propose a single domain generalization method via gradient decomposition. The author divide the gradient as domain-invariant gradient and domain-specific gradient. By this way, their approach can better solve SDG problem.

**Strengths:**

1. This paper decompose the gradient from both domain-invariant and domain-specific parts based on equilibrium principle, this idea is novelty.

2. Instead of setting conflict gradient as 0 or random value, the author set the value via domain gradient equilibrium.

3. Extensive experiments on PACS, VLCS, Office-Home and DomainNet have verified the effectiveness of proposed method.

**Limitations:**

1. What will happen if we set all w in eq.(6) as 0. Will the performance increase or decrease?

2. Have the author visualize the parameter change after their proposed gradient-based method?

3. I am confused about the experiment settings. For multi-source domain generalization, we use P, A, C, S usually represent trained model on other domains and test on P, A, C, S domain respectively. For single domain generalization, it should be P->A, P->C. so what is the results in Table 1-4, is it trained on P and average results on other 3 domains or other experimental settings?

4. Generally, ResNet-101 is more frequently used for DomainNet, could the author provide the results of DomainNet under ResNet-101 backbone?

5. The author should compare their method with more recent papers.

**Suitability:**

3

---

### Official Review · Reviewer_1prL · 2024-05-13

**Rating:** 4
**Confidence:** 4

**Summary:**

Inspired by the principles of mechanical equilibrium in physics, this paper addresses the issue of Domain Gradient Conflicts in Single Domain Generalization (SDG) by proposing Domain Gradient Equilibrium for SDG. This method decomposes conflicting gradients into domain-specific gradients and domain-invariant directions, allowing for the adjustment of domain-specific gradients' directions and weights to counteract the specific domain's influence on the overall model. The result is a model capable of generalization.

**Strengths:**

Overall, the paper is well-structured, clear, and demonstrates a certain level of innovation. The experiments are thorough and explore the issue of Gradient Conflicts comprehensively, yielding outstanding results compared to previous methods. The approach takes advantage of Gradient Conflicts and effectively disperses domain-specific gradients to neutralize irrelevant information. The experimental results offer new perspectives on addressing Gradient Conflicts.

**Limitations:**

While the Introduction elaborates on mechanical equilibrium principles, the methodology does not explicitly explain the relationship between the proposed method and these principles. It is assumed that the method aims to achieve force balance by adjusting weights, but this connection should be clarified. The paper lacks detailed explanation on how the obtained domain-invariant gradients are used to update model parameters, especially regarding the integration of gradients and parameter updates. Figure 2 only illustrates three obtained  without explaining how they are integrated for parameter updates.

Questions:

1.How are the model parameters updated using the domain-invariant gradients of each domain during training?

2.Is the direction of domain-invariant gradients in Figure 3 correct, and do they represent the decomposition direction of Domain gradients?

3.How are multiple domains projected during Gradient Adjustment? Are they projected to the same plane or multiple domains, and how is the maximization of angles between them ensured?

**Suitability:**

1

---

### Official Review · Reviewer_PEVe · 2024-05-23

**Rating:** 3
**Confidence:** 3

**Summary:**

The paper discusses a novel approach to Single Domain Generalization (SDG), where the goal is to train a model on a single source domain to generalize effectively to unseen target domains. Traditional SDG methods utilize data or feature augmentation to create diverse distributions that mimic multiple domains. However, these methods often struggle with gradient conflicts that hinder the development of domain-invariant representations.
To address this issue, the authors introduce a new technique called "domain gradient equilibrium," inspired by the concept of mechanical equilibrium in physics. This method differentiates between domain-invariant and domain-specific gradient components. By adjusting and reweighting the domain-specific gradients, the technique achieves an equilibrium that directs model optimization towards a domain-invariant orientation, thereby enhancing the model's generalization capabilities.
The effectiveness of this conflict-aware approach is validated through comprehensive experiments on four image recognition benchmarks. Notably, the method achieves a significant accuracy improvement of 2.94% on the PACS dataset compared to existing state-of-the-art approaches, underscoring the potential of this new strategy in improving SDG outcomes.

**Strengths:**

The paper propose the domain gradient equilibrium, which focus on gradient balance to advance single-domain generalization.

**Limitations:**

1. The separation of domain variant and domain specific gradient/feature is not a new thing [1].
2. Lack of comparison with the most recent methods [2].

[1] Zhang, Daoan, et al. "Aggregation of disentanglement: Reconsidering domain variations in domain generalization." arXiv preprint arXiv:2302.02350 (2023).

[2] Li, Chenming, et al. "Cross contrasting feature perturbation for domain generalization." Proceedings of the IEEE/CVF International Conference on Computer Vision. 2023.

**Suitability:**

2

---

### Meta-Review · Area_Chair_nbQb · 2024-07-07

**Recommendation:** Accept (Poster)
**Confidence:** 5

**Metareview:**

This work aims to address the single domain generalization (SDG) problem. The authors argue that most of the existing data augmentation-based SDG methods may suffer from the gradient conflict problem, which hinders domain-invariant feature learning. To address this issue, the authors proposed a Domain Gradient Equilibrium (DGE) method. DGE first decomposes the gradient into domain-specific / invariant gradient based on the divergence of each gradient vector. The domain-specific gradient will be down-weighted to achieve the Equilibrium.

The reviewers generally acknowledge the novelty of the domain-invariant/specific gradient decomposition and the experimental results on the benchmark datasets. Initially, the reviewers had some concerns about the experimental details and the comparison methods of this work. Most of the questions have been addressed in the rebuttal phase. Balancing the strengths and weaknesses of this work, I recommend accepting this paper as a poster.